

**Characterization of free amino acids, bacteria and fungi in size-**
**segregated atmospheric aerosols in boreal forest: seasonal patterns,**
**abundances and size distributions**
Aku Helin[1], Outi-Maaria Sietiö[2], Jussi Heinonsalo[2], Jaana Bäck[3], Marja-Liisa Riekkola[1] and Jevgeni
Parshintsev[1]
[1] Department of Chemistry, P.O. Box 55, FI-00014 University of Helsinki, Finland
[2] Division of Microbiology and Biotechnology, Department of Food and Environmental Sciences, University of Helsinki,
P.O. Box 56, FI-00014, Finland
[3] Department of Forest Sciences, University of Helsinki, P.O. Box 27, FI-00014, Finland
*Correspondence to: Jevgeni Parshintsev (evgeny.parshintsev@helsinki.fi)*
**Abstract**
Bioaerosols are ubiquitous in the atmosphere and constitute ~30% of atmospheric aerosol particle mass in sizes >1 µm.
Bioaerosol components, such as bacteria, fungi and pollen, may affect the climate by acting as could-active particles, thus
having an effect on cloud and precipitation formation processes. In this study, size-segregated aerosol samples (<1.0, 1-
2.5, 2.5-10 and >10 µm) were collected in boreal forest (Hyytiälä, Finland) during one year and analyzed for free amino
acids (FAAs), DNA concentration and microorganism (bacteria, *Pseudomonas* and fungi). Measurements were performed
using tandem mass spectrometry, spectrophotometry and qPCR, respectively. Meteorological parameters and statistical
analysis were used to study their atmospheric implication for results. Distinct annual patterns of bioaerosol components
were observed, late spring and autumn being seasons of dominant occurrence. Elevated abundances of FAAs and bacteria
were observed during the local pollen season, whereas fungi were observed at highest level during autumn.
Meteorological parameters, such as air and soil temperature, radiation and rainfall were observed to possess close
relationship with bioaerosol abundances on an annual scale.














## 1 Introduction

Bioaerosols are emitted directly from the biosphere into the atmosphere (Després et al., 2012;Fröhlich-Nowoisky et al., 2016). Bioaerosol are released from multiple sources, such as soil, vegetation and oceans, and they include e.g. pollen, plant fragments, spores, bacteria, algae and viruses. In recent years, the abundance and dispersal of microorganisms in the atmosphere has attracted more and more interest, mainly due to the underestimation of their abundance and their possible atmospheric impact (Jaenicke, 2005;Morris et al., 2011;Deguillaume et al., 2008;Burrows et al., 2009b;Estillore et al., 2016). It is estimated that globally bioaerosols constitute ~30% of aerosol particles mass in the particles sized >1 μm in urban and rural air (Fröhlich-Nowoisky et al., 2016). This fraction can be much higher (~80%) in the tropical forest areas (Pöschl et al., 2010;Elbert et al., 2007), and interestingly even up to 65% at boreal forest during pollen season (Manninen et al., 2014). These high percentages provide the basis for assumptions that bioaerosols may play an important role in the atmosphere by affecting cloud and precipitation formation processes by acting as cloud and ice nuclei (Huffman et al., 2013;Burrows et al., 2009b;Burrows et al., 2009a; Després et al., 2012; Fröhlich-Nowoisky et al., 2016). Thus, to clarify their atmospheric transport and ecosystem interactions, bioaerosols' chemical and microbial constituents need more characterization and identification.

Bioaerosols have been studied by using a variety of techniques (Georgakopoulos et al., 2008), but cultivation and microscopy have been frequently employed for the elucidation of microorganisms in aerosols (Manninen et al., 2014;Després et al., 2012). Nowadays, molecular genetic analysis techniques, such as quantitative polymerase chain reaction (qPCR) or next-generation sequencing (NGS), have gained ground, because they provide information not only on the viable and cultivable cells, but also on uncultivable, dead or fragments of plant and animal cells (Després et al., 2012). Alternative approach to determine particles of biological origin is based on chemical tracer techniques (Bauer et al., 2008;Gosselin et al., 2016;Hock et al., 2008;Schneider et al., 2011;Zhang et al., 2010;Staton et al., 2015;Rathnayake et al., 2017). The benefit of chemical tracers, such as carbohydrates, proteins/amino acids and lipids, is their applicability to quantitative analysis, while their main disadvantage is disability to identify different biological species. Specific tracers have been used to estimate the total amount of bioaerosols in the atmosphere (Gosselin et al., 2016;Hock et al., 2008;Schneider et al., 2011;Zhu et al., 2015).

Biologically-derived amino acids are non-volatile and hygroscopic compounds that are mainly found in the condensed phases in the atmosphere (Matos et al., 2016;Samy et al., 2013;Zhang and Anastasio, 2003). They are present in aerosols either in the combined form (proteins and peptides) or as free amino acids (FAAs), and they are emitted either from biogenic sources or formed from the degradation of proteinaceous material (Milne and Zika, 1993;Matos et al., 2016). Due to their close relationship with bioaerosols, amino acids can be used as biomarker for an overall estimation of biomass in aerosols (Hock et al., 2008;Schneider et al., 2011).

In this study size-segregated aerosol samples were collected in boreal forest during one year period. Aerosol samples were analyzed for DNA concentration, microorganism-DNA (bacteria, *Pseudomonas* and fungi) and FAAs. The annual concentration and size distribution variation of microorganisms and FAAs were investigated in detail in order to understand their potential sources. Correlation study between the bioaerosol components and meteorological parameters





was carried out. Our objective was to gain further information about the abundances of bioaerosols in different particle
size fractions at the boreal forest region and to understand better their biosphere-atmosphere interactions.

**2 Experimental section**

2.1 Materials and reagents

Detailed information on materials and reagents is presented in the Supporting Information (SI).

2.2 Aerosol sampling

The aerosol samples were collected in a Scots pine forest between February and October 2014 at the SMEAR II station
(Station for Measuring Forest Ecosystem-Atmosphere Relations) in Hyytiälä, Finland (Hari and Kulmala, 2005). A Dekati
PM10-impactor was used for the sampling of four particle size fractions (<1.0, 1-2.5, 2.5-10 and >10 µm) below the
canopy. The sampling flow rate was on average 30 L/min and the collection time was from one to four days (sampling
volume 76-144 m$^3$). The collection filters were 25 mm polycarbonate membranes (Whatman Nuclepore) for the three
largest particle size fractions. To prevent particles bouncing, membranes were smeared with diluted Apiezon L vacuum
grease. The smallest size fraction (<1.0 µm) was collected on a 47 mm Teflon filter (Gelman Sciences) with 2 µm pore
size. After sampling, the filters were dried and weighted. The sampling procedure is described in more details in Laakso
et al. (2003). After gravimetric analysis, the filters were placed inside a closed polystyrene petri dish, covered with
aluminum foil and stored at -20 °C.

Two sets of aerosol samples were collected in consecutive days, so that the sampling dates were close to each other (Table
S1). The first set of aerosol samples were analyzed for DNA and microorganisms (set A) and the other set for free amino
acids (set B). For molecular biological analysis only particles larger than 1 µm were selected because smaller fractions
were suspected to contain only virus particles and fragmented DNA.

2.3 Determination of amino acids

After ultrasonic assisted extraction, free amino acids were determined by liquid chromatography-tandem mass
spectrometry. Detailed information about the validation and quality control of the analytical method is presented in the
Supporting Information, Figure S1 and Tables S2-S4.

2.4 Extraction of DNA

Total nucleic acids were extracted from the collection filters with a commercial DNA extraction kit (PowerWater DNA
Isolation Kit, MoBio Laboratories, USA) with slight modification (see Supporting Information). The DNA concentration
and purity was measured spectrophotometrically at 260 nm and 280 nm with NanoDrop ND-1000 (Thermo Scientific,
USA). The DNA samples were stored at -20 °C prior to qPCR.




2.5 DNA amplification

The bacterial and fungal DNA amounts of the filter samples were quantified with qPCR using target specific primers pairs, Eub338F and Eub518R, and FF390 and FR1 (Table S5) (Fierer et al., 2005;Vainio and Hantula, 2000). In addition, genus specific primers, Eub338F and PseudoR, were utilized to detect the bacteria belonging to the genus *Pseudomonas* (Purohit et al., 2003).

In the bacterial and *Pseudomonas* specific qPCR reactions, standard curves were generated with DNA extracted *Pseudomonas fluorescens* H-27 (Hambi culture collection, University of Helsinki) and for the fungal specific qPCR, the DNA from the newly whole genome sequenced *Phlebia radiata* FBCC43 (genome size 40.92 Mb, FBCC culture collection, University of Helsinki) was used (Kuuskeri et al., 2016). When converting the copy number of the samples into bacterial cells (/colony forming units), all bacteria were assumed to contain average of three gene copies of ribosomal 16S DNA per cell, and bacteria belonging to genus *Pseudomonas* was assumed to contain five gene copies of 16S rDNA in a cell (Stoddard et al., 2014). According to Fröhlich-Nowoisky et al. (2009), majority of the fungal DNA detected during summer and autumn are from fungal species belonging to Basidiomycota. Based on previous results (Fröhlich-Nowoisky et al., 2012;Fröhlich-Nowoisky et al., 2009), we have assumed that most the fungal DNA collected on our filters is also derived from basidiomycete fungi. Because only few 18S rDNA-regions of fungi are fully annotated, we have selected the *P. radiata* as a model basidiomycete fungus and calculated the gene copy numbers to fungal cells based on the assumption that all fungi in this experiment have approximately the same amount of 18S rDNA gene copies as this fungus has (Kuuskeri et al., 2016).

2.6 Additional background data and back-trajectory analysis

Meteorological variables, gas fluxes and atmospheric gases are continuously measured at the SMEAR II and the data is available from AVAA-portal (Junninen et al., 2009). Half-hourly averaged data from the portal was further averaged according to each sampling time period by using arithmetic mean. These averaged values were used in statistical analyses. The selected variables were air (AT) and soil surface temperature (SST), soil surface water content (SSWC), wind speed (WS) and direction (WD), gross primary production (GPP), CO, $CO_2$ and ozone concentration, photosynthetically active radiation (PAR), UV-A and UV-B radiation and relative humidity (RH). In addition, rainfall was averaged according to different time periods as follows: 72 h before sampling (BSR), during sampling (DSR) and 168 h after sampling (ASR).

Backward air mass trajectories were calculated using the HYSPLIT (Hybrid Single-Paricle Lagrangian Integrated Trajectory) transport and dispersion model from NOAA Air Resources Laboratory to estimate the origin and transport route of aerosol particles (Draxler and Hess, 1998;Stein et al., 2015;Rolph, 2003). For the calculations, meteorological data from the Global Data Assimilation System (GDAS, 1 degree, global, 2006-present) was used. The backward air mass trajectories were modelled 48 hours back time during the whole period of sampling, using a resolution of 12-hours.

2.7 Statistical analysis



For analyzing the importance of the seasonality and the aerosol filter size, permutational multivariate analysis of variance
(PERMANOVA) was performed for the presence/absence transformed amino acid data. The PERMANOVA was
performed with the adonis-function of the vegan package (Oksanen et al., 2016) with 999 permutations, and the amino
acid data was set as response variables while sampling month and aerosol filter size were set as explanatory variables.
One-way analysis of variance (ANOVA) was performed to the fungal, bacterial and *Pseudomonas* gene copy numbers
individually with the aov-funcion from the stats package (R Core Team, 2016) in order to study the effect of seasonality
and aerosol filter size. Prior to performing ANOVA the normal distribution of each residual was checked individually
with shapiro.test-function, and all the gene copy number data was log-normalized. The linear correlations between the
FAA and microorganisms concentrations with meteorological variables were calculated using the rcorr-function of the
Hmisc package (Harrell, 2016) and visualized with the corrplot package (Wei and Simko, 2016) showing only
correlations with statistical significance ($p \leq 0.05$).
**3 Results and Discussion**
3.1 General characteristics
The average concentration and range of each component measured in different particle size fractions at the boreal forest
site is presented in Table 1 and Table S6. The lowest concentration levels of FAA and total DNA were measured during
winter and highest during late spring (Figures 1 and S2). The total amount of extracted DNA was highest from May to
June (up to 48 ng/m$^3$) and from September to October (up to 14 ng/m$^3$). The FAA concentration in total particles was
highest in late May (up to 751 ng/m$^3$) and in September (up to 35 ng/m$^3$). A common trend in both total DNA and FAA
concentrations was a maximum in spring and a secondary smaller maximum in autumn. Similarly, the highest PM
concentration in total particles was recorded during late spring and autumn (Figure S3); a pattern which has also been
previously observed at the same location (Laakso et al., 2003;Manninen et al., 2014). The measured DNA concentrations
agreed with those obtained for PM$_{2.5}$ samples at a rural mountain site in Germany, where the DNA concentration was in
the range 1.7-4.2 ng/m$^3$ (Després et al., 2007). Also the observed FAA concentrations were in the same order of magnitude
as those observed in other rural and semi-urban locations (Zhang and Anastasio, 2003;Zhang et al., 2002;Samy et al.,
2011;Samy et al., 2013). For example, at two rural sites in US the concentration of FAAs in PM$_{2.5}$ samples was measured
to be 22±9 ng/m$^3$ during summer (Samy et al., 2011), and 59±49 ng/m$^3$ (range 9-236 ng/m$^3$) during one year study period
(Zhang and Anastasio, 2003). To our knowledge, there are no studies covering DNA and FAA abundances at the boreal
forest region, thus comparison to previous results is not feasible.
Similar to the annual trend of total DNA and FAAs, the lowest concentration levels of microorganisms were detected
during winter and highest during spring and autumn. However, bacterial and fungal DNA reached their maximum levels
at different seasons; bacteria peaking in late spring (Figure 2a) and fungal DNA in late summer and autumn (Figure 3).
The bacterial DNA abundance in total particles was lower than 900 cells/m$^3$ during winter and early spring, whereas in
late spring the concentration peaked at 58731 cells/m$^3$. During summer and autumn, the concentration of bacterial DNA
was one magnitude lower. In contrast to the total number of bacteria, the highest amounts (>200 cells/m$^3$) of *Pseudomonas*
DNA were measured in late May, late June and October (Figure 2b). In rest of times, the concentration of bacteria





belonging to the genus *Pseudomonas* was much lower, and the highest concentration levels in total particles rarely
exceeded 50 cells/m$^3$. The fungal abundance was lower than 300 cells/m$^3$ during winter and early spring, whereas in late
spring and summer the concentrations started to increase and generally exceeded 6000 cells/m$^3$ in total particles. The
highest amounts of fungal DNA were measured in late June (30149 cells/m$^3$), in August (55839 cells/m$^3$) and in early
October (35050 cells/m$^3$). In general, the concentration levels of microorganisms vary seasonally and geographically, but
are estimated to be in the level of ~$10^4$-$10^5$ m$^{-3}$ and ~$10^4$-$10^5$ m$^{-3}$ for bacterial cells and fungal spores, respectively
(Burrows et al., 2009b;Després et al., 2012;Spracklen and Heald, 2014). Our observations are consistent with the common
trend, when considering that low concentration levels are typically observed at rural locations.



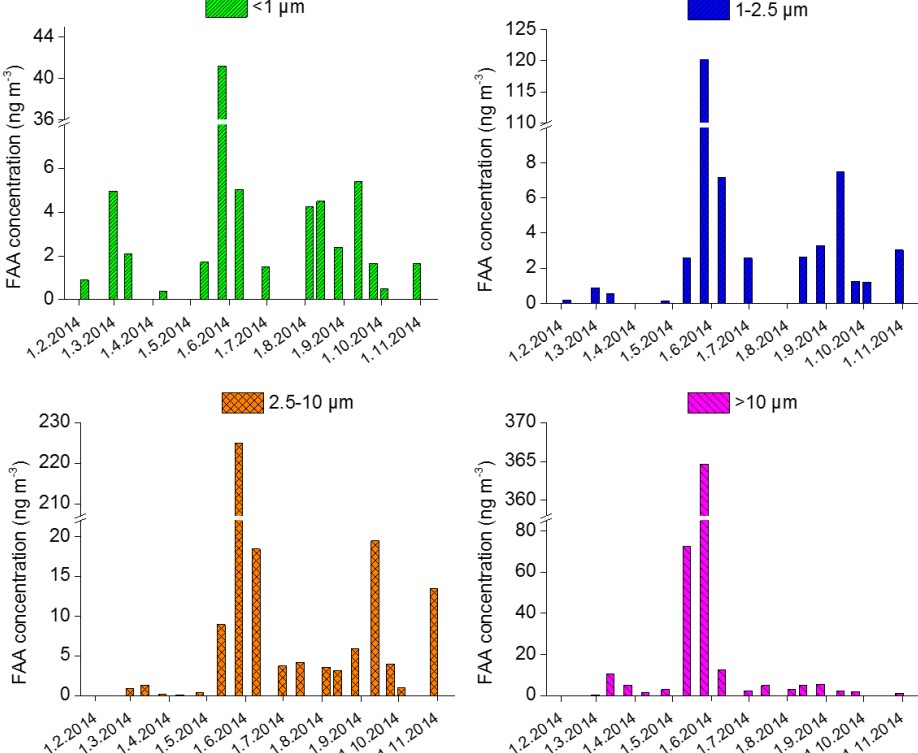


Figure 1. Free amino acid concentrations over the sampling period in different particle size fractions (date format is
dd.mm.yyyy.). Note the different y-axis scales in panels.



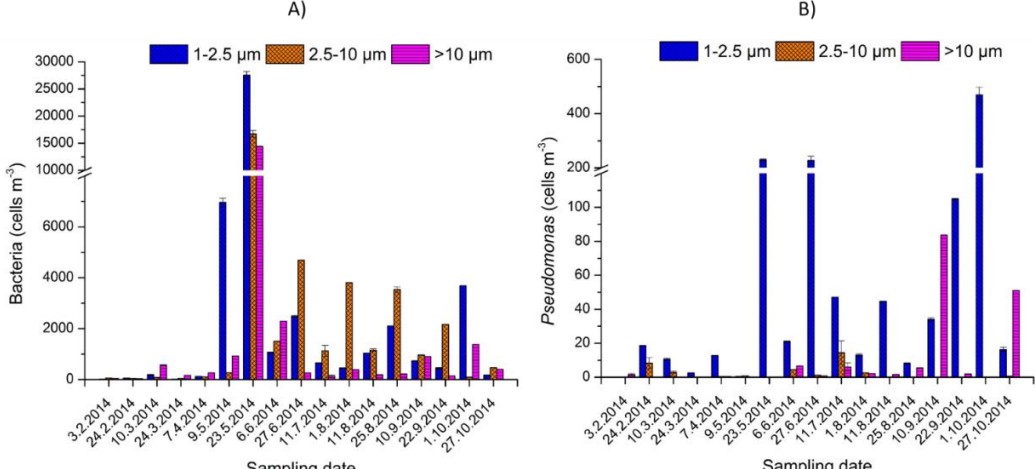


Figure 2. a) The amount of all bacterial cells over the sampling period, and b) cells of bacteria belonging to genus
*Pseudomonas*, detected with qPCR from the aerosol filters, specified by the particle size fractions (date format is
dd.mm.yyyy.).



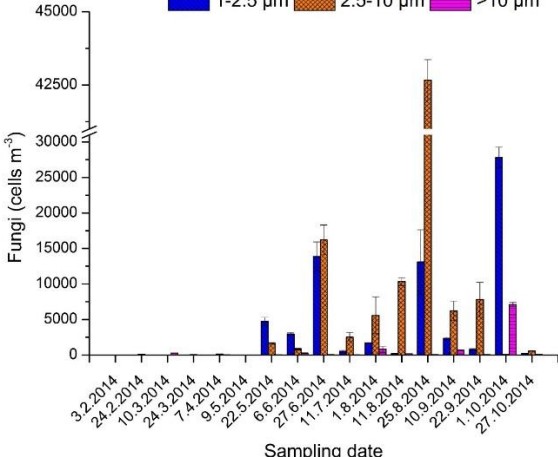


Figure 3. The amount of fungal cells and spores over the sampling period, detected with qPCR from the filters of
different particle size fractions (date format is dd.mm.yyyy.).


3.2 Seasonal variation and size distribution of FAAs

In winter the amino acids were mainly accumulated in the particle size fraction <1 µm (~74%, Figure S4a), whereas
during spring the amino acids were mainly present in the largest particle size fraction >10 µm (~77%). The highest amount
of FAAs were observed during late spring and early summer (Figure 1), when the local pollen season plumed. During
summer, the concentrations of FAAs were relatively constant and seemingly FAAs evenly distributed in all the size





fractions (Figure S4a). In autumn they were mainly present in the particles below 10 µm, and accumulated especially in
the size fraction 2.5-10 µm (~51%). During autumn, there was a slight increase in the total FAA concentrations mainly
due to the increase in the size fraction 2.5-10 µm (Figure 1).

Similarly, in the size distribution of individual amino acids some clear tendencies were observed (Figure S5). The results
from the PERMANOVA analysis indicated that seasonal variation explained the observed variation across the whole
FAA data with statistical significance (p<0.001). In addition, the size of the aerosol filter was statistically significant
factor for explaining the observed variation across the whole amino acid data (PERMANOVA, p<0.001). Glycine was
the most dominant amino acid in the size fraction <1 µm, whereas in the larger size fractions its relative abundance was
much smaller (Figure S5). On average, glycine accounted 59% of the total FAAs in particles <1 µm, followed by alanine
(10%), aspartic acid (9%) and glutamine (9%). In the size fraction 1-2.5 µm, glutamine (42%), glutamic acid (16%) and
arginine (15%) were on average the most abundant amino acids. Similarly, in the size fraction 2.5-10 µm glutamine
(29%), arginine (15%) and glutamic acid (12%) were the dominant ones, accompanied by proline (7%), serine (7%) and
alanine (7%). In the largest size fraction >10 µm proline (36%), arginine (21%), glutamine (10%) and glutamic acid (9%)
were the most dominant ones. Our results are in agreement with several studies demonstrating that glycine, arginine,
serine, alanine, proline and glutamine are the most commonly abundant FAAs in aerosols (Barbaro et al., 2011;Barbaro
et al., 2015;Samy et al., 2011;Samy et al., 2013;Scalabrin et al., 2012;Violaki and Mihalopoulos, 2010;Zhang and
Anastasio, 2003), although the relative abundances vary depending on location, season and particle size fraction (Matos
et al., 2016).

3.3 Seasonal variation and size distribution of microorganisms

A distinct monthly variation in bacterial DNA abundance in different filter size fractions was observed (ANOVA, p<0.01).
During early spring, bacterial DNA was accumulated mainly in the size fraction >10 µm (~64%, Figure S4b). Oppositely,
in late spring the highest numbers of bacteria were detected in the size fraction 1-2.5 µm, whereas during summer, highest
numbers of bacteria were discovered in the size fraction 2.5-10 µm (Figure 2a). During summer, bacteria was mainly
present in the size faction 2.5-10 µm, which covered on average ~57% of the total bacteria amount. In general, most of
the bacteria were observed in the size fractions 1-2.5 µm and 2.5-10 µm, and less bacterial cells were found in the size
fraction >10 µm. However, on average the abundances did not differ significantly in different size fractions (ANOVA,
p=0.494). The size distribution observed is in line with that of Burrows et al. (2009b) who summarized that the median
aerodynamic diameter of particles containing bacteria is 4 µm at continental sites.

The abundance of *Pseudomonas* varied with statistically significance between different months (ANOVA, p<0.05) and
between different size fractions (ANOVA, p<0.001). *Pseudomonas* was mainly present (~70%) in the smallest particle
size fraction 1-2.5 µm throughout the year (Figures 2b and S4c). The accumulation of *Pseudomonas* in the smallest size
fraction is in line with the small aerodynamic size (<1 µm) of common *Pseudomonas* species (Chang et al., 2013;Möhler
et al., 2008;Pietsch et al., 2015). Overall, the relative amount of *Pseudomonas* from all bacteria was highest in the size
fraction 1-2.5 µm (Figure S6), being highest in late winter (28%), in early spring (15%) and in autumn (23%).
Interestingly, in February the relative amount of *Pseudomonas* DNA of the total bacterial DNA was 17% in total particles
(Figure S6).





The amount of fungal DNA detected in different size fractions varied with statistical significance (ANOVA, p<0.05). The
fungal DNA was predominantly observed in the particle size fraction 2.5-10 µm, which accounted on average ~58% of
the total fungal concentration. The monthly variation in fungal DNA abundance was also statistically significant
(ANOVA, p<0.001). During May and early June, the fungal DNA was mainly accumulated in the size fraction 1-2.5 µm
(~67%, Figure S4d). In late summer, fungal DNA was detected at its highest level in the size fraction 2.5-10 µm (Figure
3), and in overall ~66% was accumulated in this size fraction during the summer months. Fungal spores are frequently
observed in aerosols in the size range of 2-10 µm (Després et al., 2012). Our observations are consistent with these
literature values.

3.4 Overview of meteorological factors and sources

The effect of local meteorological factors on the concentration levels of FAAs and microorganism was studied by means
of linear regression analysis and a summary of results is shown in Figure 4. Our results reflect rather long-term seasonal
effects (more details in Supporting Information), due to the time resolution of sampling. Thus, as expected, the key
meteorological factors explaining the observed concentration levels were air temperature, soil temperature, gross primary
production and radiation (Figures S7 and S8). In general, we believe that the positive correlation observed between FAAs
and microorganism abundances with the before mentioned meteorological variables are closely related to growing season
and seasonality, rather than being a decisive effect. These observations are in good agreement with previously reported
tendencies covering microorganism and different meteorological factors (Jones and Harrison, 2004;Burrows et al.,
2009b;Lighthart, 2000). Soil and vegetation have been previously suggested to be the predominant sources of
microorganisms in the atmosphere (Bowers et al., 2013), and our correlation results confirm these sources as discussed
below.

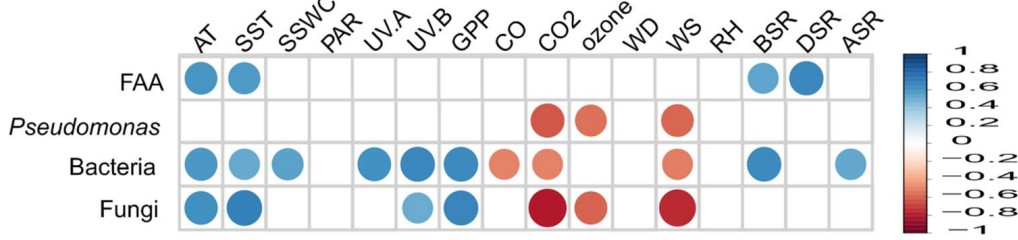


Figure 4. Summary of Spearman correlation results (total particles, p<0.05). The colour scale indicates positive/negative
correlation. Abbreviations: AT-air temperature; SST-soil surface temperature; SSWC-soil surface water content; WS-
wind speed; WD-wind direction; GPP-gross primary production; CO; CO2; ozone concentration; PAR-photosynthetically
active radiation; UV-A and UV-B radiation; RH-relative humidity; BSR-rainfall 72 h before sampling; DSR-rainfall
during sampling; ASR-rainfall 168 h after sampling.

As could be expected (Manninen et al., 2014;Schumacher et al., 2013), the lowest concentration levels of FAAs and
microorganisms were detected during winter, when the air and soil temperatures were below 0 °C and the ground was
covered by snow. In February, particles below 2.5 µm accounted for 89% of the total FAA concentration. In these samples,
glycine and alanine were the dominant amino acids. Glycine and alanine have been exploited as markers for long-range





transport aerosols due to their relatively low reactivity and long half-lives in the atmospheric condensed phases (Barbaro
et al., 2011;Barbaro et al., 2015;Scalabrin et al., 2012;Samy et al., 2013). As the concentration levels of microorganisms
were low during winter (Figures 2 and 3), it is likely that the observed FAAs originated from distant sources. However,
FAAs and particularly glycine have been also associated with biomass burning emissions (Samy et al., 2013;Violaki and
Mihalopoulos, 2010;Zangrando et al., 2016). In Finland wood-burning, that is typical domestic heating system in winter,
is the most presumable  source of over one third of the $PM_{2.5}$ emissions (Laakso et al., 2003;Saarnio et al., 2012). The
relative abundance of glycine in the size fraction 1-2.5 µm was 85% during winter, whereas in other seasons its
contribution was negligible. Thus, the observed glycine concentrations might be associated with wood-burning emissions,
coming from either local or distant sources. This conclusion is supported by the observed positive correlation between
glycine with CO concentration in the size fractions <1 µm (R=0.54, p<0.05, Figure S8) and 1-2.5 µm (R=0.80, p<0.001,
Figure S8). Furthermore, only glycine among the FAAs showed positive correlation with PM concentration in the size
fraction 1-2.5 µm (R=0.69, p<0.01, data not shown).
FAA and bacteria concentrations increased during the spring recovery and seemed to be influenced by the increase in
local biological activity. During early spring, FAAs and bacteria were mainly present in the largest size fractions (Figures
S4a and S4b), which may indicate the presence of pollen in aerosols, since microorganisms might be present on the
surface of pollen grains (Puc, 2003). Further, Manninen et al. (2014) observed pollen to be present at SMEAR II station
already before the local pollen season, most possible due to long-range transport from central Europe, where growing
season had already started. Interestingly though, during our sampling periods the back-trajectory analysis results showed
air-masses arriving primarily from Scandinavia and North Europe instead of central Europe. This could imply that the
larger sized particles are originating locally from some early pollinator rather than being long-range transported. In
addition, we observed an interesting phenomenon in early March when all FAAs, bacterial DNA and fungal DNA were
present mainly in the size fraction >10 µm. Then the corresponding relative abundances of their total amounts were 73%,
68% and 92%, respectively. During March sampling, the air temperature was almost constantly above zero degrees for
the first time in our campaign. The small increase observed in FAA and microorganism concentrations during this time
might indicate that a burst in bioaerosols emissions can occur when the temperature increases and snow starts to melt
revealing the ground vegetation and decomposed leaf litter for the first time after winter. Similar observations have been
previously made at the same site by Schumacher et al. (2013) in a study covering fluorescent biological aerosol particles
(FBAP), although they detected an increase in FBAP concentrations in November after first snowfall and snowmelt event.
The concentrations of FAAs and bacteria reached their maximum levels during the local pollen season peak in May
(Figures 1 and 2). Elevated amino acid concentrations have been observed during spring time also in other locations and
suspected to be influenced by pollen (Barbaro et al., 2011;Zhang et al., 2002). In our study, especially proline and arginine
concentrations increased during the pollen season peak. These amino acids have been shown to contribute significantly
to the total amino acid content of birch pollen (Ozler et al., 2009). In this study, the FAA concentrations increased in all
the size fractions (Figure 1), possibly due to rupture of relatively large (~30 µm) pollen grains (Taylor et al., 2004;Visez
et al., 2015), supported by visibly yellow impactor plates in all the size fractions. During the peak in FAA and bacteria
concentrations, the air-masses were arriving in Hyytiälä from Eastern Europe and Baltic Sea. Due to the absence of
pollinating species in the sea region, the observed high concentration levels of FAAs and bacteria could be mainly
explained by local sources in the boreal forest. This is corroborated by the presence of methionine, cysteine and tryptophan





only in these aerosols samples. Particularly these amino acids are known to be highly reactive with short half-lives in the
atmospheric condensed phases (Scalabrin et al., 2012;Milne and Zika, 1993;McGregor and Anastasio, 2001). Thus, the
local pollen season likely explains our observations, although other factors may partly contribute. For example, high
ozone concentration and strong UV radiation were recorded during this time period, and in some studies ozone has been
demonstrated to promote the decomposition of protein and peptides into free amino acids increasing the FAA
concentrations (Samy et al., 2013;Mumford et al., 1972). Positive correlation was observed between FAA concentration
in the size fraction >10 µm with ozone (R=0.58, p<0.05, Figure S8), which was mainly attributed to coinciding peaks in
concentration levels during pollen season.

Opposite to FAA concentrations being highest in the largest size fractions, bacterial DNA abundance was highest in the
size fraction 1-2.5 µm during the pollen season peak. As spring proceeds, new foliage growth enables larger surface area
for epiphytic bacteria to occupy and grow on. Bacterial cells may be lifted from leaves into the air on pollen (Jones and
Harrison, 2004), which may explain the peak in bacterial abundances during this season. Bacteria may be present in the
air as individual cells, clump of cells or attached to other particles, such as pollen grains and leaf fragments. We propose
that during the local pollen season and under favourable meteorological conditions, it is possible to observe a significant
increase in bacterial concentrations. However, most likely the magnitude of this increase varies from year-to-year and is
also closely related to varying pollen abundances.

Although we assume that the elevated bacterial DNA abundances are mainly related to pollen and vegetation during
spring, other sources might contribute as well. For example, soil water content and bacteria concentration correlated
positively in the size fractions 1-2.5 µm (R=0.53, p<0.05, Figure S7) and >10 µm (R=0.57, p<0.05, Figure S7). After
snow melts, the soil moisture increase enhances the bacterial/microbial growth (Burrows et al., 2009b). When spring
proceeds and air and soil temperatures increase, the relatively dry soil surface layer might enable the dispersal of bacteria
via soil resuspension. Similarly, some studies indicate that soil-derived bacteria dominate during spring time (Rathnayake
et al., 2017), while others indicate that soil sources dominate during late summer and fall (Bowers et al., 2013). In addition,
rainfall may promote the bacterial growth on vegetation surfaces, leading to increased population sizes, which may
become airborne following rainfall (Bigg et al., 2015). We observed positive correlation between bacteria concentration
in the size fraction 1-2.5 µm with rainfall recorded 72 h before the sampling (R=0.64, p<0.01, Figure S7). Based on
previous reports related to the relationship between rainfall and bioaerosols (Huffman et al., 2013;Prenni et al.,
2013;Gosselin et al., 2016;Rathnayake et al., 2017;Morris et al., 2016;Bigg et al., 2015), our results corroborate the
positive effect of bacteria abundances following rainfall.

The abundance and relative size distribution of fungal DNA varied seasonally and started to increase in summer (Figure
3). The relative distribution of fungal DNA to different size fractions correlated with the relative humidity (R=-0.53,
*P*=0.035 for 1-2.5 µm and R=0.45, *P*=0.081 for 2.5-10 µm, data not shown). The lower the relative humidity was, the
more fungi were accumulated in the smallest size fraction. Since fungal cells are typically larger than 2.5 µm, the observed
fungal DNA in the 1-2.5 µm fractions during early summer is probably originated from the spores of moulds or ruptured
cells (Reponen et al., 2001). In addition, the size of fungal spores has been found to depend on the relative humidity, i.e.
higher the relative humidity is the larger the spores are (Reponen et al., 1996;Madsen, 2012). The spores of basidiomycete
fungi are in general larger than those of ascomycete (Reponen et al., 2001;Manninen et al., 2014;Hussein et al.,




2013;Fröhlich-Nowoisky et al., 2012), and the basidiomycete fungi are known to sporulate mainly during autumn when
the relative humidity is high (Kauserud et al., 2011). Further, in the previous study by Manninen et al. (2014) the spores
of Basidiomycota dominated the phylum-level distribution of fungal spores in the autumn. Our results where fungal DNA
accumulated in the 2.5-10 µm size fraction during autumn are in agreement with this study. Consequently, the annual size
distribution of fungi can be expected to be similar from year-to-year in the boreal forest.

Opposite to the trend of fungi, FAA and bacterial DNA concentrations decreased after the spring pollen season peak. It
is noteworthy that both bacterial and fungal DNA were accumulated during the mid-summer to early fall in the particle
size fraction 2.5-10 µm (Figure S4b and S4d). In summer, the air masses arrived Hyytiälä mainly from Scandinavia and
Baltic Sea regions. In early August the air masses reached Hyytiälä from a large forest fire region in Sweden. Interestingly
derived from our samples, glycine was detected in relatively high concentrations (~2.9 ng/m$^3$) in the size fraction <1 µm.
This observation supports the hypothesis that the presence of glycine is partially related to biomass burning emissions
and long-range transport. However, no distinct effects on the origin of air-masses were seen in the levels of FAAs or
microorganisms. Although long-range transport cannot be completely ruled out, the accumulation of bacteria and fungi
in the particle size fraction 2.5-10 µm suggest that primarily local forest or nearby sources affect the concentration levels
of microorganisms during summer. For example, leaf-associated bacteria are known to contribute to the total bacterial
amount during summer (Bowers et al., 2013). Statistically significant correlation was observed between gross primary
production (R=0.79, P<0.001, Figure S7) and photosynthetically active radiation (R=0.69, P<0.01, Figure S7) with
bacteria in the size fraction 2.5-10 µm, possibly indicating the importance of vegetation especially during summer and
growing season.

The abundance of *Pseudomonas* was detected at its highest level in early October in the size fraction 1-2.5 µm, when it
was 2-fold higher than in other months (Figure 2b). It is suggested that an increase in the *Pseudomonas syringae*
population size on vegetation may occur during or after rainfall (Hirano et al., 1996;Bigg et al., 2015;Morris et al., 2016).
Even though no correlation was observed between average rainfall recorded before or during sampling with *Pseudomonas*
on an annual scale, we identified interesting separate individual rainfall events. The maximum single rainfall event was
recorded within 72 h before the above mentioned early October sampling (Figure S9). As far as we could tell, this was
the main exception in meteorological conditions during the period studied, i.e. the only factor potentially explaining the
observed increase in *Pseudomonas* concentration. Further, the back-trajectory analysis results showed air-masses arriving
from Norwegian Sea and North Sea via the Scandinavian Mountains, and to our knowledge these areas do not contain
sources that could explain the increase in *Pseudomonas* concentration. When considering the other maximum rainfall
events recorded before sampling, two out of three of the summer-autumn maximum events coincided with high
*Pseudomonas* concentrations (Figure S9). These findings corroborate the evidence of a potential relationship between
rainfall and *Pseudomonas*. However, the lack of statistically significant correlation suggests that other factors are
contributing to observed variation in *Pseudomonas* abundance.

Overall, considering the effect of rainfall on the levels of bacteria, fungi and FAAs, some interesting patterns were
observed. As mentioned earlier, the bacterial DNA concentration in size fraction 1-2.5 µm correlated with rainfall
recorded prior to sampling. Further, we observed positive correlation between FAA concentration and rainfall during
sampling in the size fractions 1-2.5 µm (R=0.53, p=0.0502, data not shown) and 2.5-10 µm (R=0.70, p<0.01, Figure S8).



Interestingly, bacterial DNA and fungal DNA abundances correlated positively in the size fraction 2.5-10 µm with rainfall
recorded 168 h after sampling (R=0.63, p<0.01 and R=0.53, p<0.05, respectively; Figure S7). There is relatively strong
evidence that cloud-active particles larger than ~1 µm are biological in origin (Haga et al., 2014;Hassett et al., 2015;Mason
et al., 2016;Möhler et al., 2007), and in mixed-phase cloud conditions, bioaerosols may play an important role in triggering
rainfall. Collectively, the different correlations observed between rainfall and bioaerosol components suggest that a
potential feedback mechanism may persist at the boreal forest. This conclusion is based on the assumption that the
recorded rainfall events after sampling were produced at least partially on a local scale and that the meteorological factors
were favourable to formation of rainfall. We acknowledge, that the positive correlation observed between
microorganism's abundances with rainfall recorded following sampling, may be causal in nature. However, in light of the
recent findings, the possibility of a feedback mechanisms cannot be ignored (Bigg et al., 2015;Morris et al., 2016;Huffman
et al., 2013). Nonetheless, additional high time resolution and long-term measurements are needed to confirm the
observations presented in this research.

**4 Conclusions**

Considering the observations made in this and previous studies (Manninen et al., 2014;Schumacher et al., 2013), some
general conclusions related to bioaerosol abundances and size distribution at the boreal forest site can be drawn.
Correlation with meteorological parameters might indicate biosphere-atmosphere interactions through bioaerosols with
possible climate effects. In early spring, bioaerosol components are mainly accumulated in the size-fractions >10 µm.
According to our results, the spring pollen season has an impact on the pollen levels as well as on the bacterial abundances.
Elevated bioaerosol abundances occur during the pollen season (Manninen et al., 2014), and based on our estimation even
up to ~77% of total PM may be of biological origin (SI). Thus, the magnitude of biological cloud-active particles during
this period may be atmospherically relevant (Diehl et al., 2002;Diehl et al., 2001;Pummer et al., 2012;Pummer et al.,
2015). The importance of rainfall was observed in this study as well as to some extent in previous studies conducted at
the same site (Manninen et al., 2014;Schumacher et al., 2013). Positive correlation was observed between bioaerosol
component abundances with rainfall recorded before and during sampling as well as with rainfall recorded after sampling.
During late summer and autumn, the accumulation of bioaerosols in the size fraction 2.5-10 µm was evident (Figure S4).
In autumn the relative amount of biomass in PM was estimated to be around ~10% (SI). Modelling studies have suggested
that microorganisms may play an important role in the hydrological cycle in the boreal region(Sesartic et al., 2012;Sesartic
et al., 2013). However, the magnitude of upward lifting of microorganisms remains to be solved, and thus our results are
preliminary in nature and need to be confirmed.



**Author contribution**
A.Helin, O.-M. Sietiö, J. Heinonsalo, J. Bäck, M.-L. Riekkola, J. Parshintsev designed the experiments. A. Helin, O.-M.
Sietiö, J. Parshintsev carried them out. A. Helin and O.-M. Sietiö performed the statistical analysis. A. Helin and J.
Parshintsev prepared the manuscript with contributions from all co-authors.
**Competing interests**



The authors declare that they have no conflict of interest.
**Acknowledgements**
The financial support of the Academy of Finland Center of Excellence program (project no 272041, JP, MLR, AH) and
research project no 292699 (OMS, JH) are gratefully acknowledged. Technical staff of the SMEAR II station are thanked
for their valuable help. Magnus Ehrnrooth foundation (JP) and University of Helsinki Doctoral Program in Microbiology
and Biotechnology (MBDP) (OMS) are thanked for support. Geoffroy Duporté is acknowledged for back-trajectory
analysis. Merck Life Science is thanked for providing ZIC-cHILIC columns.

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

Table 1. Average (±standard deviation) concentration and range of each component measured and the number of filter
samples (n) analyzed in total.

| Component | <1 µm | 1-2.5 µm | 2.5-10 µm | >10 µm | total particles |
|---|---|---|---|---|---|
| PM (µg/m³) | 5.9±4.1 | 2.1±2.3 | 2.4±3.2 | 2.1±3.5 | 12.3±12.2 |
| n=138 | (1.0-18.1) | (0.2-11.9) | (0.2-15.9) | (0.03-17.4) | (2.2-63.3) |
| FAA (ng/m³) | 5.22±10.11 | 10.95±31.54 | 18.45±53.55 | 27.62±85.71 | 57.91±174.17 |
| n=69 | (<LOQ-41.21) | (<LOQ-120.24) | (<LOQ-224.92) | (0.04-364.65) | (1.13-751.01) |
| DNA (ng/m³) | - | 5.16±5.20 | 1.99±3.47 | 2.18±3.76 | 8.60±11.41 |
| n=51 | | (<LOQ-18.56) | (0.002-14.67) | (<LOQ-14.80) | (0.24-48.04) |
| Bacteria (cells/m³) | - | 2811±6619 | 2171±4032 | 1341±3424 | 6323±13748 |
| n=51 | | (17-27551) | (56-16746) | (41-14434) | (137-58731) |
| *Pseudomonas* (cells/m³) | - | 74±125 | 4±5 | 13±26 | 86±122 |
| n=51 | | (0.1-469) | (<LOQ-14) | (<LOQ-84) | (1-469) |
| Fungi (cells/m³) | - | 4022±7518 | 5579±10614 | 648±1809 | 10173±15622 |
| n=51 | | (2-27838) | (9-42667) | (<LOQ-7129) | (27-55839) |