# Peer review of "segregated atmospheric aerosols in boreal forest: seasonal patterns,"

_Atmospheric Chemistry and Physics, 2017_

## Referee Comment (RC1) · Anonymous Referee #1 · 16 Aug 2017

General comments: In this work, the authors study the concentration and particle size distribution variation of twenty free amino acids (FAAs), DNA concentration and the DNA concentration of specific microorganism (bacteria, Pseudomonas and fungi) in size-segregated aerosol samples collected in a boreal forest (Hyytiälä, Finland) between February and October to understand their potential source. For this purpose, two sets of aerosol samples were collected in consecutive days: the first set of aerosol samples were analysed for DNA and microorganisms-DNA and the second set for FAAs. The FAAs were determined by liquid chromatography-tandem mass spectrom-

etry, the DNA was extracted with a commercial DNA extraction kit and the concentration and purity was measured spectrophotometrically, and specific bacterial and fungal DNA were quantified with quantitative polymerase chain reaction (qPCR). In order to study their atmospheric implication, the authors also assessed the statistical strength of the linear correlations between FAA and microorganisms concentrations with meteorological parameters (e.g., air temperature, soil surface temperature, soil surface water content, wind direction and speed. Permutational multivariate analysis of variance (PERMANOVA) was also performed to evaluate the changes in the concentration of different FAAs according the season and the aerosol filter size. In my opinion this article is relatively well written and it is well within the scope of the Atmospheric Chemistry and Physics journal. The analytical techniques used and the analytes studied (FAAs and DNA concentrations), are not often associated, which allows for a deeper understanding of the concentration of primary biological particles in atmospheric aerosols. Furthermore, from an environmental point of view it is extremely relevant to know in detail the contribution of these biological particles to the atmospheric aerosol pool to identify their potential sources and the biosphere-atmosphere interactions. Moreover, I would also like to highlight the quality of the Supporting Information. The supporting information contains a very exhaustive set of data with details both on materials and reagents as well as validation procedures and results associated with the determination of amino acids. The results are well within the expected for a fit for purpose method and namely the recovery experiments and the assessment of expanded measurement uncertainty supports the concern of the authors for obtaining high standards for the analytical control quality. Finally, I would like to emphasize that, taking into account the correlation study, one cannot assert that this the statistics used, provide a strong support to the conclusions drawn, particularly since these linear correlations normally range between 0.5 and 0.7 (positive and negative) and consequently will be "moderate" correlations, the most. Nevertheless, since the authors often use several references to support their claims, the study turns out to be well substantiated. Considering my opinion above, I believe that this study should be accepted in the Atmospheric Chem-

istry and Physics journal. Nevertheless, below I present some suggestions and notes I would like to see clarified.

Specific comments: In page 2 line 42, the authors start the introduction with the statement "Bioaerosols are emitted directly from the biosphere into the atmosphere (Després et al., 2012;Fröhlich-Nowoisky et al., 2016)". Although this statement is not conceptually wrong, one of the cited articles, i.e. Després et al., 2012 (Després, V.R., Alex Huffman, J., Burrows, S.M., Hoose, C., Safatov, A.S., Buryak, G., Fröhlich-Nowoisky, J., Elbert, W., Andreae, M.O., Pöschl, U., Jaenicke, R., 2012. Primary biological aerosol particles in the atmosphere: a review. Tellus B 64, 1–58. doi:10.3402/tellusb.v64i0.15598), recommends the use of the term " primary biological aerosol particles (PBAP)" instead of "bioaerosol". Therefore, I suggest the authors to check the definition presented Després et al., 2012 and consider its revision throughout the article.

Throughout the manuscript, the authors mention several times that the sampling period was one year. However, in section 2.2, the sampling period was defined from February to October. Although cover all seasons, there are 3 months missing and the sampling period is in fact only 9 months. This information should be corrected in the manuscript.

Line 203. The bacterial cells and fungal spores have the exact same concentration levels, or there is some mistake in this sentence?

Line 339. The presence of particles enriched with FAAs from the sea bubble-bursting phenomena?

Line 342. Barbaro et al., (Barbaro, E., Zangrando, R., Vecchiato, M., Piazza, R., Cairns, W. R. L., Capodaglio, G., Barbante, C. and Gambaro, A.: Free amino acids in Antarctic aerosol: potential markers for the evolution and fate of marine aerosol, Atmos. Chem. Phys., 15(10), 5457–5469, doi:10.5194/acp-15-5457-2015, 2015) introduces an argument contradictory to that presented in this manuscript. According to Barbaro et al., (2015), the enrichment of aerosol samples in hydrophobic FAAs (e.g., methionine, cysteine and tryptophan) supports the assumption that long-range transport processes, as the different chemical and photochemical events that occur during long-range transport were faster for hydrophilic than for hydrophobic amino acids. Can authors comment on this contradiction?

In line 423, the authors state that "there is relatively strong evidence that cloud-active particles larger than $\sim$1 $\mu$m are biological in origin". In my opinion, to be a "strong evidence" needs to be better justified.

In conclusion, after reading the Supporting Information, I believe that I understood the estimations made by the authors to reach the percentage of PM that should be of biological origin. However, since the two conversion factors used are subject to high uncertainties, in my opinion, these estimation uncertainties should be emphasized in the text of the manuscript, to avoid misleading the reader. In Supporting Information, section "Validation experiments and quality control" the authors state that "Most of the amino acid calibration curves were forced through origin". Could you please justify this choice?

---

## Referee Comment (RC2) · Anonymous Referee #2 · 23 Aug 2017

**General comments**

In this study, DNA and free amino acids (FAA) were analyzed using quantitative polymerase chain reaction (qPCR) and liquid chromatography-tandem mass spectrometry techniques, respectively, in order to characterize particulate matter of biological origin (bioaerosols). In qPCR, two target specific amplicons and one genus specific amplicon were chosen to quantify bacterial and fungal DNA and DNA of the genus Pseudomonas. For each sampling interval covering several days, two sets of size segre-

gated (<1.0, 1-2.5, 2.5-10 and > 10μm) aerosol samples were collected consecutively for the two analysis methods in a boreal forest at the SMEAR II station in Hyytiälä, Finland. Data sets ranging from February to October 2014 were complemented with meteorological and atmospheric gas data from the same site. FAA and microorganisms concentrations derived from qPCR results were analyzed for correlation to meteorological data and their seasonal trends interpreted together with backward air mass trajectories calculated with the HYSPLIT model from NOAA Air Resources Laboratory. FAA and DNA showed high abundances during the spring pollinating season and in autumn corresponding to concentration maxima of bacteria and fungi, respectively. Their distribution over size fractions provided information on contributing dispersion mechanisms. Also rainfall was shown to correlate to bioaerosol abundances.

The analysis methods proposed in this paper represent a substantial progress in the ability to track sources of biological aerosol particles as target groups or genus specific abundances. Linear correlations from (multivariate) variance analysis of FAA and DNA abundances also supported interpreting interactions of aerosol sources with meterological parameters. The manuscript outlines appropriately background information on bioaerosols including an extensive list of references followed by the experimental section which is kept compact and fluent by moving details to the Supporting Information (SI). Assumptions and estimations necessary for quantification such as the calculation of cell concentrations are described clearly and backed by references. The extensive SI includes detailed descriptions not only of the sample set collected as well as the materials and analysis procedures used, but also information on testing and validation measures taken to verify critical parameters affecting data quality such as blanks, recovery and selectivity. Results and interpretations are described clearly, taking into account related work. Also, limitations of the current work and an outlook on future research exploiting higher time resolution and long term measurements are pointed out. Considering the above, I therefore recommend the manuscript to be accepted subject to a few technical corrections.

[Figure]

**Specific comments**

As already pointed out by referee #1, the use of the term primary biological aerosol particles (PBAPs) defined in detail in one of the references (Despres et al., 2012) may be considered instead of bioaerosols. At the end of page S21 in the supplementary information it is already in use without prior definition.

**Technical corrections**

**line 97 word 9:** weighted → weighed

**line 121 end:** primers pairs → primer pairs

**line 586:** Ozler → Özler

―――――――――――――――――――

---

## Author Comment (AC1) · 20 Sep 2017

We appreciate a lot the work done by the reviewers and wish to express our gratitude to their critical comments. The manuscript definitely benefits from the suggested corrections. Below, we give the detailed answers to the reviewer's suggestions, while text is corrected with track changes in the revised manuscript.

Referee 1

(1) Bioaerosols are emitted directly from the biosphere into the atmosphere (Després et

al., 2012;Fröhlich-Nowoisky et al., 2016)". Although this statement is not conceptually wrong, one of the cited articles, i.e. Després et al., 2012 (Després, V.R., Alex Huffman, J., Burrows, S.M., Hoose, C., Safatov, A.S., Buryak, G., Fröhlich-Nowoisky, J., Elbert, W., Andreae, M.O., Pöschl, U., Jaenicke, R., 2012. Pri- mary biological aerosol particles in the atmosphere: a review. Tellus B 64, 1–. doi:10.3402/tellusb.v64i0.15598), recommends the use of the term " primary biological aerosol particles (PBAP)" instead of ". Therefore, I suggest the authors to check the definition presented Després et al., 2012 and consider its revision throughout the article.

(2) Both terms are used in the literature, but we agree with the Referee, that PBAP suits better here.

(3) Terms "Bioaerosols" was corrected to "Primary biological aerosol particles" and abbreviated as PBAP throughout the article.

(1) Throughout the manuscript, the authors mention several times that the sampling period was one year. However, in section 2.2, the sampling period was defined from February to October. Although cover all seasons, there are 3 months missing and the sampling period is in fact only 9 months. This information should be corrected in the manuscript.

(2) We agree.

(3) Corrected throughout the article.

(1) Line 203. The bacterial cells and fungal spores have the exact same concentration levels, or there is some mistake in this sentence?

(2) We reported a wide range of typical concentration levels for both fungal spores and bacterial cells based on the references cited in the text. Therefore, the values pre- sented seem to be identical or in the same order of magnitude, and there is no mistake in the reported concentration ranges. However, we acknowledge that in reality, the concentration levels are seldom similar as mentioned in Line 202. The concentration

levels vary depending on the geographical location, meteorological factors etc. Nevertheless, in our opinion, adding additional comments on varying microorganisms' levels is not relevant at this point.

(3) No changes were made to the manuscript.

(1) Line 339. The presence of particles enriched with FAAs from the sea bubble-bursting phenomena?

(2) In principle, it might be possible that there is some enrichment due to bubble-bursting phenomena, although this is not likely the case here. This was not the only sampling period during which the air-masses were passing the Baltic Sea region (e.g. during non-frozen sea periods, Line 391). When taking into account the presented evidence that supports the contribution of pollen, e.g. pragmatically considering that the filters were visibly yellow, it is not convenient to speculate that the vast peak in concentration levels would be caused by bubble-bursting phenomena.

(3) No changes were made or comments added to the manuscript.

(1) Line 342. Barbaro et al., (Barbaro, E., Zangrando, R., Vecchiato, M., Piazza, R., Cairns, W. R. L., Capodaglio, G., Barbante, C. and Gambaro, A.: Free amino acids in Antarctic aerosol: potential markers for the evolution and fate of marine aerosol, Atmos. Chem. Phys., 15(10), 5457–, doi:10.5194/acp-15-5457-2015, 2015) introduces an argument contradictory to that presented in this manuscript. According to Barbaro et al., (2015), the enrichment of aerosol samples in hydrophobic FAAs (e.g., methionine, cysteine and tryptophan) supports the assumption that long-range transport processes, as the different chemical and photochemical events that occur during long-range transport were faster for hydrophilic than for hydrophobic amino acids. Can authors comment on this contradiction?

(2) As far as we could interpret, there is no such contradiction present between our results and the ones presented by Barbaro et al. In their "hydropathy" index classification, only Met is included in hydrophobic amino acids, whereas Trp and Cys are not accounted for. Nowhere in their article is a statement that the presence of Met, Cys and Trp would be indicative of long-range transported aerosols. Oppositely, in another article by the same authors (Scalabrin et al., ACP 2012, doi: 10.5194/acp-12-10453-2012), they emphasize that Met is not typically observed in long-range transported aerosols. We considered that there is enough references in the current version of the manuscript to support our observations.

(3) No changes were made or comments added to the manuscript.

(1) In line 423, the authors state that is relatively strong evidence that cloud-active particles larger than 1 $\mu$m are biological in origin". In my opinion, to be a "strong evidence" needs to be better justified.

(2) We agree that evidence might not be strong.

(3) "Strong" changed to "some".

(1) In conclusion, after reading the Supporting Information, I believe that I understood the estimations made by the authors to reach the percentage of PM that should be of biological origin. However, since the two conversion factors used are subject to high uncertainties, in my opinion, these estimation uncertainties should be emphasized in the text of the manuscript, to avoid misleading the reader.

(2) We agree with the referee. However, detailed explanation on estimation uncertainty does not suit the "conclusion" part. Thus, we emphasized the SI and added "high uncertainty" to the text.

(3) The following sentences were modified: "Elevated PBAP abundances occur during the pollen season (Manninen et al., 2014), and based on our estimation even up to $\sim$77% of total PM may be of biological origin (see SI for details). Even though our estimation is highly uncertain, the magnitude of biological cloud-active particles during this period may be atmospherically relevant (Diehl et al., 2002;Diehl et al., 2001;Pummer

et al., 2012;Pummer et al., 2015)."

(1) In Supporting Information, section "Validation experiments and quality control "the authors state that of the amino acid calibration curves were forced through origin". Could you please justify this choice?

(2) The relevance of forcing through the zero was not dramatic on the results due to intercepts being close to zero anyway. However, in practice it was noticed that forcing through zero gave more realistic LOQ value, i.e. higher values than compared to not forcing through zero.

(3) No changes were made to the manuscript

Referee 2

(1) As already pointed out by referee #1, the use of the term primary biological aerosol particles (PBAPs) defined in detail in one of the references (Despres et al., 2012) may be considered instead of bioaerosols. At the end of page S21 in the supplementary information it is already in use without prior definition.

(2) We agree.

(3) Corrected throughout the article.

(1) line97word9:weighted → weighed

(2) We agree.

(3) Corrected.

(1) line121end:primers pairs → primer pairs

(2) We agree.

(3) Corrected.

(1) line586:Ozler → Özler

(2) We agree.

(3) Corrected.
* * *

---

## Author Response (AR2)

Author's response

Dear Editor,

thank you for noticing! Authors were added to the reference and p-values were added to the Figure 4.